# An Analysis of the Supply of Open Government Data

**Alan Ponce [1],\* and Raul Alberto Ponce Rodriguez [2]**

1   Institute of Engineering and Technology, Autonomous University of Cd Juarez (UACJ),
    Cd Juárez 32315, Mexico
2   Institute of Social Sciences and Administration, Autonomous University of Cd Juarez (UACJ),
    Cd Juárez 32315, Mexico; rponce@uacj.mx
\*   Correspondence: alan.ponce@uacj.mx

**Abstract:** An index of the release of open government data, published in 2016 by the Open Knowledge Foundation, shows that there is significant variability in the country's supply of this public good. What explains these cross-country differences? Adopting an interdisciplinary approach based on data science and economic theory, we developed the following research workflow. First, we gather, clean, and merge different datasets released by institutions such as the Open Knowledge Foundation, World Bank, United Nations, World Economic Forum, Transparency International, Economist Intelligence Unit, and International Telecommunication Union. Then, we conduct feature extraction and variable selection founded on economic domain knowledge. Next, we perform several linear regression models, testing whether cross-country differences in the supply of open government data can be explained by differences in the country's economic, social, and institutional structures. Our analysis provides evidence that the country's civil liberties, government transparency, quality of democracy, efficiency of government intervention, economies of scale in the provision of public goods, and the size of the economy are statistically significant to explain the cross-country differences in the supply of open government data. Our analysis also suggests that political participation, sociodemographic characteristics, and demographic and global income distribution dummies do not help to explain the country's supply of open government data. In summary, we show that cross-country differences in governance, social institutions, and the size of the economy can explain the global distribution of open government data.

**Keywords:** data science; open government data; governance and social institutions; economic determinants of open data

## 1. Introduction

Open data (OD) refers to information that has been generated by public or private entities and then published under a license that allows its use, reuse, and distribution freely [1]. Information collected and released from the public sector (e.g., transportation, pollution, agriculture, education, health, and census, among others) is referred to as Open Government Data (OGD) [2]. The public sector is considered one of the main contributors to the open data movement, due to the vast amount of information it generates [3]. According to [4], during recent years, there has been an increase in the number of countries that are adopting open data policies as part of their governmental agenda. Authors also argue that this trend is related to the potential benefits that OGD offers as a shared value (social and economic). From the social perspective, OGD is considered as a trigger of transparency, accountability, fighting against corruption, and the empowerment of citizens. The economic aspect of OGD is related to fostering innovation, enterprise opportunities, and job creation, because OGD is considered a production asset in the digital economy.

Additional evidence of global interest in the open data topic is the recent creation of different portals in which governments consolidate their data from different public entities (e.g., education, health, transportation) on a single website in order to release their data for free and collective use. Some examples of these portals developed by governments are the US (https://www.data.gov/open-gov/), Canada (https://open.canada.ca/en/open-data), Brazil (http://www.dados.gov.br/), Mexico (https://datos.gob.mx/), or the European Data Portal (https://www.europeandataportal.eu/en) funded by the European Commission. Other aspects related to open data interests are the initiatives constituted in conjunction with citizens, academics, and non-governmental organizations that are creating indexes, such as the Global Open Data Index (GODI) (https://index.okfn.org/), Open Data Barometer (ODB) (https://opendatabarometer.org/?_year=2017&indicator=ODB), Open Data Watch (ODW) (https://opendatawatch.com/), and Open Data Impact (ODI) (https://odimpact.org/) which are measuring the amount of data published by different governments around the world, as well as potential benefits and challenges (technical, legal, economic, social) that these public datasets (e.g., education, health, transportation) are generating in society.

Although there has been an increased interest in the phenomenon of open government data, most research has been conducted by applying qualitative methodologies through surveys, case studies, and desk research focusing on diverse topics, such as challenges and barriers in adopting and implementing open government data initiatives, and other qualitative studies have been focused on the release, provision, or value of these public datasets [5–7]. However, there is a gap in the literature for analyzing and measuring the determinants of the supply of open government data adopting a quantitative approach. This work pretends to fill this gap and contribute to the state of the art of open government data, providing a statistical analysis explaining countries' variabilities of the release of open government data through economic, social, and institutional factors. According to the Global Open Data Index published in 2016 by the Open Knowledge Foundation (OKF) (https://okfn.org/), there are significant differences across countries in the supply of open government data. In particular, Australia, the United Kingdom, and France obtain the highest GODI scores reported by the Open Knowledge Foundation (meaning that these countries contribute the most to the supply of open government data), while countries such as Myanmar, Barbados, Malawi, and Botswana obtained the lowest records on the GODI score (meaning that, in a global comparison, these countries contribute the least to the supply of open government data). This leads us to the following question: What explains the high heterogeneity in the global supply of open government data?

The objective of this paper is to provide an answer to this question by extending a single academic perspective, due to this research being based on an interdisciplinary approach aligned by the fields of data science and economics. The intersection point of these disciplines involves analyzing and estimating the determinants of the heterogeneity in the supply of global open government data by means of gathering information from different sources, featuring extraction and variable selection, modelling through the implementation of statistical methods, and explaining the effect and relationship of this heterogeneity. On the one hand, the data science approach is implemented in order to systematically create a data pipeline collected from different portals. This task is executed following a process of obtaining, scrubbing, exploring, modelling, and interpreting (OSEMN) the information collected from several sources. Then, we apply feature engineering in order to extract and analyze the data by way of a regression model that seeks to analyze the statistical association between some political and economic determinants of open government data. To estimate our model of regression analysis, we develop a sample with country cross-section data with data of the Global Open Data Index (GODI) for the year 2016. In this process, we solve empirical issues that arise in the regression analysis, such as multicollinearity, heteroscedasticity, missing data, outliers, and high dimensionality with our target variable (open government data).

On the other hand, economic theory is adopted to develop an empirical analysis (using our data pipeline) for the analysis of variables and their justification based on domain knowledge. Open government data is considered as a pure public good [8]; that is to say, we consider open

government data as satisfying two important properties: it is a non-excludable (once open government data is provided, then any person who seeks access can have access to that good) and it is a non-rival good (the consumption of open government data by some agent does not preclude the consumption of the same good by everyone else). Applying this theoretical framework, we test if political and social institutions such as civil rights, transparency, quality of democracy, and political participation, as well as economic and sociodemographic characteristics at the country level (such as the size of the economy, the efficiency of the government, the demand for Internet services, the median age of the population of a country, and the size of the population), can explain the global variability in the supply of open government data.

Using a cross-country regression model, our analysis provides evidence that cross-country differences in governance and social institutions such as civil liberties, government transparency, and the quality of democracy are statistically significant predictors of cross-country differences in the supply of open government data. Our estimates suggest that the government's transparency and civil liberties have a marginal positive and statistically significant effect on the supply of open government data. In our model, our variable that captures changes in the demand of web resources, that being the penetration of users (the proportion of Internet users over the country's population), is also positively and statistically significant in all of our estimated models.

In addition, our indicators of the efficiency of government intervention and economies of scale in the provision of public goods (analyzed through the variable population in each country) are also statistically significant predictors of cross-country differences in open government data. Our models also provide weak support to the hypothesis that open government data is a normal good; that is to say, countries with higher incomes are associated with higher levels of supply of open government data. Finally, our estimates suggest that political participation, the sociodemographic characteristics of citizens, demographic dummy variables, and dummy variables capturing the global distribution of income do not help to explain cross-country differences of the supply of open government data. In summary, we find evidence that cross-country differences in the supply of open government data are associated with the heterogeneity of social and political institutions, and economic factors are also correlated with the supply of open government data.

It is relevant to mention that the main limitation of our analysis is that we use cross-section data for our regression analysis, which limits the generality of our results. We decided to use data from the GODI for the year 2016 because this is the most up-to-date data on the GODI. Even if there is data for the Global Open Data Index for other years, the Open Knowledge Foundation has clearly stated that changes in methodology in the calculation of the GODI make unsuitable the comparison of data between 2016 and other years. This limits the study of what factors could explain the changes of the GODI over time. However, this limitation could be eased, as long as more data sets become available in the future that allow other forms of regression analysis, such as regression with panel data, that might improve the properties of estimation and hypothesis testing, as well as the generality of the results.

The structure of the paper is as follows. Section 2 includes a brief literature review, postulating the technical, social, economic, and political determinants of global open government data. Section 3 describes the data collection, the preparation process for our analysis, and the identification of the linear regression model. Section 4 contains the results of our analysis. Section 5 concludes the paper.

## 2. Literature Review

The adoption and implementation of open data is a socio-technological phenomenon that has been studied by different disciplines, trying to understand and estimate its dimensions and barriers [9–11]. For instance, the technical outlook is associated with the relevance of improving the data interoperability, quality, accessibility, usability, accuracy, platforms, and infrastructure needed in order to release open data [12–17]. The social stance refers to the empowerment that data offers to society such as, for example, the potential benefits that the information released by governments could produce through transparency and the accountability of citizens [18–21]. The economic point of view is related to possible impacts on

the economy that open data could offer through the creation of new businesses, products, and services, as well as employment [22–24]. This perspective also includes the crucial role that innovation plays as a driver of economic growth in the private and public sectors using open data [25–30]. The political perspective covers the strategies, policies, and impacts of the data released by the state [31–34]. The data published and freely accessible by public entities is referred to as Open Government Data (OGD). This particular kind of data plays an important role in the open data movement because it is considered as one of the main supporters through legislations such as the Open Data Directive (https://ec.europa.eu/digital-single-market/en/european-legislation-reuse-public-sector-information) or global political initiatives like the Open Government Partnership (OGP) (https://www.opengovpartnership.org/). These political actions aim at increasing efficiency, promoting transparency, empowering citizens, and driving a knowledge-based economy through the release of data generated by public sectors. The study performed in [35] claims that the creation of open data policies is essential for defining the financial and technological infrastructure required, publication process definition, legal framework certainty, and political sustainability of open government data (OGD). The author also argues that open data policies should disseminate the economic and social value of OGD in order to stimulate the use and reuse of it in society. It is argued in [36] that the release of OGD is relevant because there are datasets collected by different sectors and for specific purposes (e.g., transportation, pollution, agriculture, education, health, and census). The authors also claim that OGD is a driver for innovation and business opportunities for society. Finally, they argue that the infrastructure of these data sets is paid by taxpayers; therefore, this information is considered a public good.

*2.1. Determinants of the Supply of Open Government Data*

In this section, we develop an analysis of the determinants of the supply of open government data. Hence, we explain the incentives of government policy makers to provide goods and services. As we mentioned before, in this paper, we consider open government data as a pure public good which has two important properties: it is a non-excludable (once a pure public good is provided, then any person who seeks to have access can have access to that good) and non-rival good (the consumption of the good by some agent does not preclude the consumption of the same good by everyone else). In our analysis, we consider that households and firms demand open government data because they find value in it; that is to say, households and firms might find open government data valuable because this information might help them to make rational and informed decisions. This information can also be used to foster their objectives, such as engaging in civic activities, political debates, and other activities regarded as desirable. For the case of households and in the case of firms, open government data might help them to make more efficient decisions (see [22,37,38]).

The literature on public economics has made important contributions to the study of the provision of this type of good, and this literature can be classified in two distinctive lines of research. The first line is the normative theory, and the second is the positive theory of the provision of public goods. The normative literature has emphasized that the preferences of households for private and public goods, the technology of production, and the costs of taxation that finance these goods are the main determinants of the provision of public goods (for a comprehensive review of the normative literature on public goods, see [39] and, more recently, [40]).

In contrast, the positive literature on public goods has emphasized that, in addition to household preferences and the technology of production, governments are suppliers of public goods, and candidates to public office are elected through a democratic process. It should be pointed out that the supply of open data has a production cost and, therefore, governments need to allocate public budgets for the collection and administration of data. This means that the allocation of budgets that allow the supply of open government data is subjected to a regular process of political negotiation in Congress and executive powers. Therefore, the provision of public goods could be explained by electoral incentives; political candidates seek to win public office, and they compete for votes in an election to form the government and make decisions over public policy (see [41,42]). For this reason,

politicians have incentives to provide public goods that benefit a significant proportion of voters in the electorate, with the hope of attracting votes in the election and maintaining political support while politicians hold office.

To be more specific about how electoral competition creates incentives for politicians to provide different levels of goods and services, we describe in detail the quid pro quo of models of electoral competition (see [41,42]). In a democracy, voters with different socio-demographic characteristics such as age, gender, and income might demand certain goods and services from the government because they benefit from these goods and services. Hence, candidates might consider that the distribution of demands of voters for goods and services from the government might be characterized by Figure 1. For the purpose of exposition, we assume $g$ is the size of the provision of the public good. Hence, Figure 1 shows that there might be voters who would like the lowest size of the public good, equal to *gmin* (maybe because he or she does not benefit from the provision of this good), while *gmax* is the size of the provision of other voters who want the highest level of $g$ in the distribution (maybe because the personal characteristics of these voters make them benefit a great deal from this good). Every point in the line shown in Figure 1 represents the ideal policy demanded by a certain voter, and the position *gMV* is the ideal policy demanded by the median voter; that is to say, this position is the voter who is in the middle of the distribution of policies demanded by all voters participating in the election.

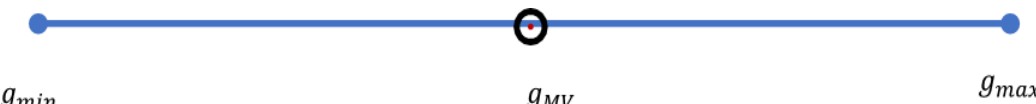

**Figure 1.** Distribution of policies demanded by voters.

Models of electoral competition consider that a voter will vote for the party that provides the policy that is closer to the voter's own preferences over policies. To see this, assume two parties, say parties 1 and 2, are competing for the vote of a particular voter, with the ideal policy given by *gh* (shown in Figure 2). This voter will vote for party 1 because the policy position of this party is closest to the voter's own preferences for this public good or service.

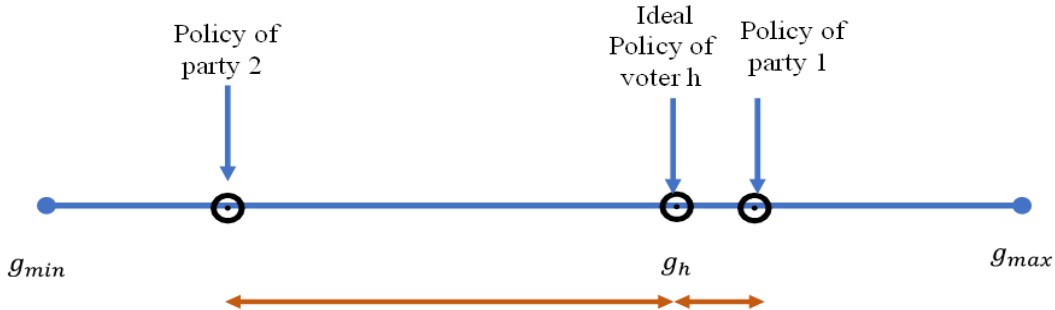

**Figure 2.** Policy positions of parties and the choice of the vote.

Hence, models of electoral competition predict that parties who want to maximize the expected votes to be received in the election should decide to offer the provision of the public good or service demanded (or desired) by the median voter (see Figure 3). That is, the government should select a level of its policy equal to $g = gMV$. By doing so, the expected proportion of the vote for each party is 50% of the vote. If any party deviates from providing the median voter policy, then the party expects to receive a proportion of the vote lower than 50% of participating voters and will lose the election. Hence, models of electoral competition make a strong prediction that can be tested empirically: parties who want to maximize the electoral support from voters in the election should estimate the demand for goods and services of the median voter.

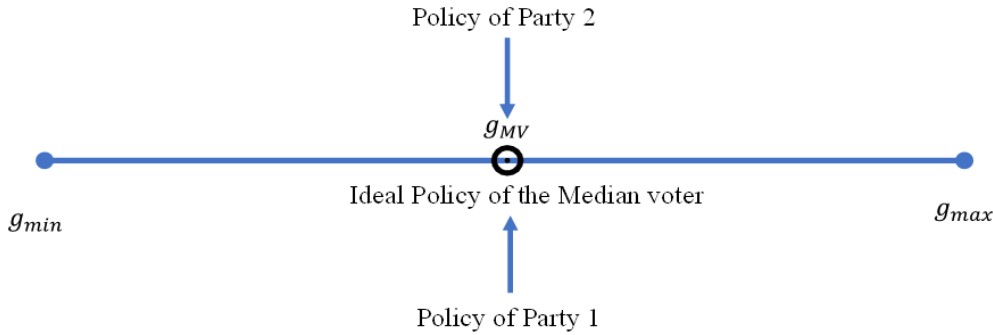

**Figure 3.** Prediction of models of electoral competition.

A further example of how this mechanism works is the following. Assume the demand of the median voter for goods and services from the government increases (perhaps because the median voter has more income and desires more government services). Then, policy makers in the government should increase the supply of government services to satisfy the demand of the median voter. It is relevant to mention that there is a great deal of evidence suggesting that public goods, such as education, health, and infrastructure (roads and bridges), which are provided by national and subnational governments, are correlated with the incentives of elections and political competition. For global empirical evidence of such a relationship covering 118 countries for three decades, see [43].

The positive literature on public goods has emphasized that, if elections matter and if there is perfect electoral competition (as a similar concept to the idea of perfect economic competition), then parties select the ideal provision of public goods of the median voter [42]. In this case, there is electoral accountability, meaning that elections create incentives for the provision of public goods that satisfy at least a majority of voters. However, if there is imperfect electoral competition and parties have preferences for public goods (that is to say, individuals controlling parties desire certain types and levels of public goods), then parties might select the ideal policy of activists inside parties or the ideal policy over public goods of a minority group of voters in the electorate (In an election, there could be imperfect electoral competition when a party does not have strong incentives to use in its policy positions to attract a majority of the votes in the election. This could be the case if voters do not vote for parties based on their policy positions, but instead on party identification (whether a voter self-identifies with a party), or the choice of the vote could be strongly determined by other non-policy issues, such as the personal characteristics of candidates (e.g., age, gender). For instance, if a significant proportion of voters (for purposes of exposition, let us say 30% of voters) vote for some party based on party identification, then this party does not necessarily select the policy of the median voter, because this party only needs another 21% of the vote to win the election. In this case, the policy positions of parties might be heavily influenced by the preferences, rather than the policies, of candidates or influential groups of voters inside of the party. Hence, there might be low electoral accountability and, if there is an increase in the demand of voters for open government data, the government might not respond by changing the supply of open government data. In this case, the demand of voters for that public good might not be satisfied) (see [44]).

In this latter case, there might be little electoral accountability, and the provision of public goods might be different relative to the ideal policy of the median voter in the economy (which might be considered as the ideal public policy for the society as a whole). Hence, the quality of the democratic process matters to determine the degree of electoral accountability and the size of the provision of public goods. Hence, for democracies with electoral accountability, if there is an increase in the demand of voters for open government data, well-functioning governments might increase the supply of open government data to satisfy the demand of voters for that public good (see [45–48]).

The demand for public goods can also be related with political participation. Citizens express their demand for public goods and services through voting in elections (see [40,42,49]). The political

participation of citizens can be observed by different political channels, such as voting in elections, attending meetings to express their demands for specific goods and services to elected representatives in congress and executive powers, or by contributing to political campaigns. Hence, we expect that more political participation leads to more accountability and better governance in democracies. Therefore, in democracies in which there is electoral accountability, higher political participation should lead to a better match between the public goods and services demanded by citizens and the supply of such services by the government.

A well-functioning democracy is also related with the civil liberties of citizens and the provision of public goods (for analysis along these lines, see [50]). Civil liberties are associated with the access of free printed and electronic media which provides relevant information to all citizens. Civil liberties can also be related with freedom of association and protest and, more relevant to our analysis, with political institutions that foster the free access to the Internet. Hence, we expect that more civil liberties are positively associated with less political restrictions to access the Internet and, therefore, more demand of content freely available on the Internet. In this line of thinking, the supply of open government data should also be positively related to transparency from the government. Transparency might help well-informed voters and economic agents to make rational decisions about the functioning of the government. Hence, voters might demand that their government provides useful information about the decisions of public policy by their elected officials. Therefore, in countries in which citizens demand more transparency, we could expect that their government satisfies this demand by providing more open government data.

The literature has also recognized that the sociodemographic characteristics of individuals, such as age, gender, and marital status, might be important determinants of the demand of public goods and services. (For a classical analysis on this issue see [51] and, for a literature review of the impact of socio-demographic characteristics on the size of government spending on public goods and other type of goods and services, see [50]) Hence, changes in the sociodemographic characteristics of households are related with changes in the demand for public goods and services (for instance, a change in the average age of individuals might lead to a change in the demand of certain services such as public education and public welfare). Therefore, changes in the sociodemographic characteristics of voters in a democracy might lead to changes in their demand for public goods, and governments have incentives to change their supply of public goods accordingly.

Most theoretical models that seek to explain the demand of private and pure public goods consider whether public goods are normal, neutral or inferior (see [52,53]). If a good is normal, then an increase in the income of households leads to an increase in the demand for such a good. If a good is inferior, then an increase in household income leads to a fall in the demand of such a good. When income increases, households might substitute the demand of low-quality goods for high-quality goods, which might explain why the demand of certain goods might fall as household income increases. A neutral good does not respond to changes in household income (see [54]). Hence, we could expect that an increase in the country's income might lead to an increase (or fall) in the demand of open government data if this good is normal or inferior, and governments might respond by increasing (or reducing) the supply of open government data.

In addition, most theoretical models that study the provision of pure public goods consider the size of the population as an important determinant of the provision of pure public goods (see [39,52,53]). As we mentioned before, a pure public good is non-excludable (once a pure public good is provided, then any person can have access to that good) and non-rival (the consumption of the good by some agent does not preclude the consumption of the same good by everyone else). Under these circumstances, the non-excludable property of a pure public good means that there could be economies of scale in the costs of providing a pure public good (see [55]). This means that the per capita costs of providing a pure public good decrease as the cost of public goods are shared among more people. In addition, an increase in the size of the population might also be associated with an increase in the size of the tax base that finances the provision of a pure public good (see [52,53]). This, in turn, leads to a fall

in the per capita cost of providing public goods, which increases the demand for this type of goods. Therefore, we could expect that an increase in the size of the population of a country might lead to an increase of the demand of open government data, and governments might respond by increasing the corresponding supply of such goods.

In summary, to guide the empirical analysis to be conducted in the following sections, we have relied on formal economic theory to characterize empirically verifiable tests of probabilistic determinants on the provision of open government data. In our analysis, we consider open government data as a pure public good because it satisfies two properties identified in the economic literature: the non-excludable and non-rival good properties. Based on the contributions of economic theory, we state several hypotheses about a probabilistic relationship between the supply of open government data by a country and the quality of democracy, the country's political participation, civil liberties and transparency, the sociodemographic characteristics of the country, the size of the economy, the size of the country's population, and the demand of content freely available on the Internet.

## 3. Material and Methods

### 3.1. Data Collection and Preprocessing

A common task in the economics and data science fields is the collection of datasets from different sources in order to discover knowledge, patterns, and trends. This activity represents some challenges, such as the diversity of the data structure, formats, and time consistency, among others. In this research, we adopted the OSMEN workflow methodology (shown in Figure 4), which is the acronym of obtain, scrub, explore, model, interpret, proposed by [56], to deal with these challenges. This methodology was proposed in the data science research community in order to systematically collect data and provide research transparency and result reproducibility.

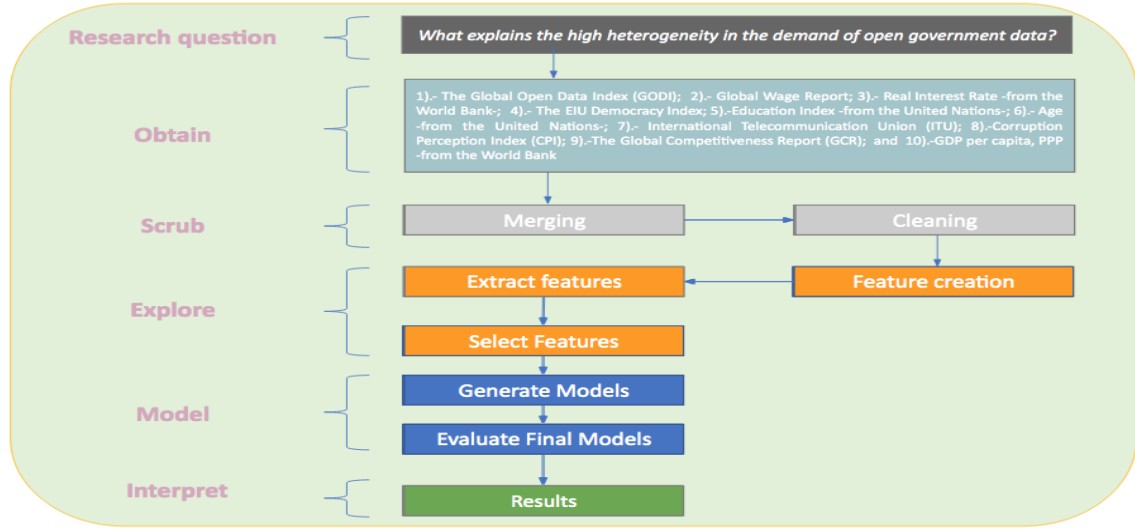

**Figure 4.** Illustration of the data science pipeline developed for our research.

Following this workflow to solve our research question, the first step was to select and gather the data from several sources, such as (1) the Global Open Data Index (GODI) from the Open Knowledge Foundation, (2) the global wage report from the International Labor Organization, (3) real interest rates from the World Bank, (4) the democracy index from the Economist Intelligence Unit, (5) the education index from the United Nations, (6) age data from the United Nations, (7) Internet use from the International Telecommunication Union, (8) the corruption perception index (CPI) from Transparency International, (9) the global competitiveness report (GCR) from the World Economic Forum, and (10) the gross domestic product (GDP) per capita and purchasing power parity (PPP) from

the World Bank. In this research, we defined the GODI indicator as our dependent variable (also called the response or target variable), and the other information extracted from these datasets became our independent variables (also called label variables).

Once the phase of data acquisition was completed, the next tasks were data integration and cleaning. The former involved analyzing and merging heterogeneous datasets. For example, some information was published in a long format and other datasets as wide formats containing different periods of time. The latter is associated with keeping consistency among datasets, such as homologating the names of countries because some datasets have different name labels (e.g., in some datasets, the country name was denoted as the United States of America or Venezuela, RB, and in other datasets, the country names appeared as the United States or Venezuela, respectively). Another aspect related to the cleaning process was to identify elements such as missing values, outliers, or other noise elements that could affect the quality of our model [57].

The next step was related to feature engineering, particularly feature creation, extraction, and selection. Some of our collected variables were categorical data. Therefore, we needed to create new variables in order to perform our models. This process is known as one-hot encoding in machine learning or dummy variables in econometrics. Then, we performed feature extraction, implementing principal component analysis (PCA), which is the process of dimensionality reduction from a large number of attributes without losing meaningful information by removing redundant data. This reduction process helped to identify features that could be more conducive to our analysis [58]. After performing PCA, our analysis indicated that there was multicollinearity in our independent variables. The topic of multicollinearity is an ongoing research area in feature engineering due to its implications when using diverse datasets [59–61]. For this reason, we include in the next section a variable selection process and a robustness check, based on an economic domain knowledge approach [62] and justified on the theoretical background introduced in the literature [63]. In order to complete a full sample with the desired variables for our empirical analysis, our final cross-sectional dataset was constituted by 18 variables and 49 observations during the year 2016. In the next section, we describe our model generation, interpretation, and results.

*3.2. Empirical Analysis*

In this section, we test if political and social institutions such as civil rights, transparency, quality of democracy, and political participation, as well as economic and sociodemographic characteristics at the country level (such as the size of the economy, the efficiency of the government, the demand for Internet services, the median age of the population of a country, and the size of the population), can explain the global variability in the supply of open government data. To test our hypotheses, we used one of the most popular tools in data science and economics: a linear regression analysis. This allowed us to estimate the marginal effect of how changes in independent variables (such as the size of the economy of a country, the sociodemographic characteristics of individuals in a country, civil liberties, and transparency) affect the supply of open government data. In the next model (see Equation (1)), we postulate that cross-country differences in the supply of open government data are associated with political and economic factors that affect the quality of democracy and the incentives of governments to provide open government data:

$$Od_i = \alpha + \beta'X + \varepsilon_i \tag{1}$$

In Equation (1), the differences in the supply of open government data across countries $Od_i$ for $i = 1, \ldots I$, where the sub-index distinguishes the different countries in our sample, is explained by a set of $k$ independent variables contained in the vector $X$ (such as political participation, the size of the economy, socio-demographic characteristics of households, and indicators of demand for Internet services). The vector $\beta\prime = [\beta_1, \beta_2 \ldots \ldots \beta_k]$ represents the marginal effect of exogenous changes in $X_i$ in our indicator of the supply of open government data; that is to say, $\frac{\partial Od_i}{\partial X_i} = \beta_i$. Our model also allowed us to test whether the marginal effect of $X_i$ on open government data was statistically significant or not.

Finally, $\varepsilon_i$ is a random error term from our model. To be more specific, the model we tested in our analysis is specified as follows:

$$Od_i = \alpha + \beta_1 GdpPPP_i + \beta_2 Dem_i + \beta_3 PolPar_i + \beta_4\, Liberty_i + \beta_5 Trans_i + \beta_6 Pop_i$$
$$+\beta_7 Age_i + \beta_8 GovEfficiency_i + \beta_9 IntPen_i + \varepsilon_i \tag{2}$$

Hence, in Equation (2), our model tests the postulated determinants of the supply of open government data discussed in Section 2.1 of this paper. Therefore, we were interested in testing whether the supply of open government data is associated with changes in the economic size of the country (see $GdpPPP_i$, which is the purchasing power parity of the gross domestic product of a country and affects the demand of open government data and its corresponding supply). The effect of the quality of democracy of a country is defined as $Dem_i$, which is a metric that measures the function, state, and trust of political freedoms and civil liberties through pillars such as political participation of citizens (defined as $PolPar_i$), civil liberties (defined as $Liberty_i$), and the transparency of a country's government (see $Trans_i$).

Other determinants in our model are the size of the population (defined as $Pop_i$) and the sociodemographic characteristics in a country (characterized by the median age of the population of a country, $Age_i$). Besides that, there is the efficiency of the government (labelled as $GovEfficiency_i$), which is an index that measures and compares per country the burden of government regulation, legal framework performance, and transparency of public policies, and our indicator of demand for Internet services (defined by $IntPen_i$, or Internet penetration), which is the number of Internet users as a proportion of the population in each country. To estimate our model of regression analysis, we developed a sample with country cross-section data. Our variable for open government data is the global open government data index (defined as $Od_i$), published by the Open Knowledge Foundation (OKF), which provides cross-country differences on the supply of open government data.

To estimate the model in Equation (2), we used a cross-section analysis with data on the GODI for the year 2016. A well-identified regression analysis needs to consider the possibility of endogeneity, which might bias the estimates of the model. Endogeneity might arise when changes in the independent variables $X$ might be correlated with changes in the dependent variable $Y$, and changes in $Y$ might also lead to changes in the $X$ variables. In this case, the marginal effects in the regression model in (2) would not be properly identified. A standard way to solve this issue is to use the independent variables $X$, but lagged for one period (for technical analysis on this issue, see [64]). Hence, to avoid the possibility of endogeneity in our estimates, we used data for the year 2015 for our control variables; that is, we used the lagged observation for the control variables, such as the size of the country, the quality of democracy, political participation, civil liberties, transparency, the size of the population, the sociodemographic characteristics in a country, and efficiency of the government. In this case, changes in $X$ could be correlated with changes in the dependent variable $Y$, but not the opposite case.

In summary, our assessment criteria for our empirical analysis was constituted as follows. First, we used theoretical analysis from the literature on economics on the main determinants of public goods and services provided by governments to identify control variables of the regression analysis (as a way to determine the structure of the $X$ variables in the regression analysis, seen in Section 2.1). The theoretical analysis provided a rationale for a probabilistic link between the independent variable (GODI) and the explanatory variables of the model in Equation (2). Second, we used standard techniques of regression analysis to determine the best way to obtain unbiased and efficient estimators of the marginal effects of the independent variables over the dependent variable GODI. Third, we conducted a robustness test of our estimates and hypothesis testing by estimating several models (see Models I, II, III, IV and V in Table 1 in the following section) to test whether our results were sensitive to specific forms of linear regression analysis.

**Table 1.** Results of OLS estimators with heteroscedasticity-consistent standard errors.

| Variable | Supply of Open Government Data GODI Score (I) | Supply of Open Government Data GODI Score (II) | Supply of Open Government Data GODI Score (III) | Supply of Open Government Data GODI Score (IV) | Supply of Open Government Data GODI Score (V) |
|---|---|---|---|---|---|
| C | 11.04 | 50.17 | 55.5180 | 53.3683 | 55.9864 |
| Gdppp | 0.0002 * | 0.0002 | 0.0002 | 0.0002 | 0.0002 |
| | (1.6998) | (1.5876) | (1.6380) | (1.3166) | (1.4044) |
| Liberty | 0.4111 ** | 0.4974 *** | 0.5252 ** | 0.4306 * | 0.4679 * |
| | 2.1968 | 2.5259 | 2.4246 | 1.7019 | 1.6761 |
| Dem | −0.4726 | −1.2298 * | −1.3926 * | −1.4452 * | −1.4923 * |
| | −1.1856 | −1.8287 | −1.6874 | −1.8960 | −1.6831 |
| Polpar | 0.1581 | 0.1925 | 0.1978 | 0.2730 | 0.2410 |
| | 0.7461 | 0.8889 | 0.8105 | 1.1275 | 0.8072 |
| Age | −0.3381 | −0.3287 | −0.2821 | 0.0522 | 0.0645 |
| | -0.9542 | −0.9782 | −0.7732 | 0.1132 | 0.1178 |
| Transparency | 15.8476 * | 17.48105 ** | 16.1369 * | 13.9994 | 14.4925 |
| | 1.7610 | 1.9572 | 1.7938 | 1.5910 | 1.6067 |
| Efficiency | −19.6248 ** | −22.12 ** | −20.8876 ** | −18.6995 * | −19.6564 * |
| | −2.0313 | −2.2669 | −2.1968 | −1.8295 | −1.94 |
| Population | 2.5681 *** | 3.1520 *** | 3.3581 *** | 3.3492 *** | 3.3951 *** |
| | 3.4713 | 3.50 | 3.3188 | 3.0274 | 2.8543 |
| Internet-Penetration | 0.5345 *** | −0.0142 | −0.1968 | −0.2213 | −0.2840 |
| | 2.9659 | −0.0474 | −0.4547 | −0.4952 | −0.5341 |
| Dem*Internet-Penetration | | 0.0088 * | 0.0114 * | 0.0116 * | 0.0129 * |
| | | 1.8122 | 1.8913 | 1.8565 | 1.8966 |
| High income | | | −0.015 | | −2.8034 |

**Table 1.** *Cont.*

| Variable | Supply of Open Government Data GODI Score (I) | Supply of Open Government Data GODI Score (II) | Supply of Open Government Data GODI Score (III) | Supply of Open Government Data GODI Score (IV) | Supply of Open Government Data GODI Score (V) |
|---|---|---|---|---|---|
| | | | −0.0009 | | −0.1298 |
| Upper Middle Income | | | 4.8105 | | 1.6383 |
| | | | 0.3667 | | 0.1074 |
| Lower Middle Income | | | 0.6255 | | −0.0153 |
| | | | 0.0842 | | −0.0016 |
| East Asia Pacific | | | | 3.9374 | 2.9713 |
| | | | | 0.3864 | 0.2726 |
| Europe Central Asia | | | | −1.2709 | −2.5741 |
| | | | | −0.1179 | −0.2231 |
| Latin America Caribbean | | | | 8.0636 | 5.0072 |
| | | | | 0.6578 | 0.3605 |
| Middle East North Africa | | | | 0.8853 | 2.6890 |
| | | | | 0.0724 | 0.2168 |
| North America | | | | 2.0636 | 0.7462 |
| | | | | 0.1751 | 0.0595 |
| | | | | | |
| Adjusted R-squared | 0.6387 | 0.6605 | 0.6689 | 0.6792 | 0.6824 |
| F-statistic | 7.66 *** | 7.39 *** | 5.43 *** | 4.65 *** | 3.58 *** |
| Sample | 49 | 49 | 49 | 49 | 49 |

*** $p < 0.01$, ** $p < 0.05$, * $p < 0.10$. All tests are two-tailed, and *t*-tests are below the corresponding estimates (numbers in parenthesis correspond to the *t*-test).

## 4. Results

We estimated our model with a set of different independent variables for a robustness check of our analysis. Our estimation technique used ordinary least squares with heteroscedasticity-consistent standard errors [65–68]. This technique allows having credible estimates of the probability distribution functions of the marginal effects $\beta_i$ associated with the independent variable $X_i$ and, therefore, credible hypothesis testing. Table 1 shows our empirical results, with each column considering different econometric specifications. Model I used our basic set of explanatory variables described in Equation (2). Model II incorporated an interaction term between democracy and Internet penetration, which allowed us to test if the quality of democracy leads to a differentiated response of countries to an increase in the demand of Internet services. Model III expanded Model II by incorporating geographical dummies. Model IV incorporated dummies related to the world's distribution of income, and Model V incorporated geographical and global income distribution dummies.

Our estimates showed that cross-country differences in governance and social institutions, such as civil liberties, the quality of democracy, and the degree of government transparency, are statistically significant predictors of cross-country differences in the supply of open government data. To see this (as shown in Table 1), note that all models show that the marginal effect of liberty on the supply of open government data is positive and statistically significant (When we refer to the marginal effects of a change of one variable and the variable of the GODI, this should be interpreted as a probabilistic marginal effect that the increase of one variable is correlated with increases or reductions, depending on the sign of the coefficient, the variable of the GODI. Hence, our analysis does not show causality, but a probabilistic correlation) (at different levels of value $p$). In addition, the government's transparency also has a marginal positive and statistically significant effect on the supply of open government data in Models I, II, and III, while the coefficient of the quality of democracy is statistically significant in Models II, III, IV, and V (see Table 1), and the interaction term between democracy and the penetration of users of the Internet is positive and statistically significant in all models in which we considered this interaction term.

In addition, our variable that captured changes in the demand of web resources—that is, the variable of penetration of users in our model, or the proportion of Internet users over the country's population—is positively and statistically significant in all of our estimated models. In Models II, III, IV, and V, we included an interaction term to test whether differences in the quality of democracy lead to different responses by governments to changes in the demand of use of the Internet, termed Dem*Penetration. The marginal effect of the interaction term is positive and statistically significant in all of our models in which we used this variable. That is to say, all countries in the sample have a marginal positive response in the supply of open government data when they observe increases in the demand of Internet services, but countries with higher qualities of democracy supply more open government data than countries with weaker democracies. This result confirms that governance is an important determinant of the cross-country differences in the supply of open government data.

However, in our models, political participation and the sociodemographic characteristic of citizens (in our models, the average age of citizens) were not statistically significant in any of the estimated models. The marginal effect of political participation had a positive effect (as expected) while the average age of citizens had a negative effect (as intuition might suggest) in Models I, II, and III, but a positive effect in Models IV and V. In addition, our models provided, at best, weak support to the hypothesis that open government data is a normal good; that is to say, countries with higher incomes are associated with higher levels of supply of open government data, since the positive income effect on open government data is significant only in Model I. Once we included demographic dummy variables and dummy variables of the global distribution of income, the marginal effect of the size of the economy on the supply of open government data was positive, but not statistically significant (see Models II, III, IV, and V).

In our analysis, two variables were associated with the efficiency of government intervention (this being the variable efficiency in Table 1) and economies of scale in the provision of public goods,

analyzed through the variable population. Our estimates showed that the efficiency of the government of a country has a negative and statistically significant marginal effect on the supply of open government data in all of our models. One possible explanation of this outcome is that an increase in the efficiency of government intervention might lead to more resources available to be spent by governments. However, those available resources are spent on other governmental programs that might have a high electoral impact relative to the choice of supplying more open government data. Therefore, a more efficient government increases spending in some programs (with high electoral impacts) and reduces spending in other programs (with relatively low electoral impacts). In addition, the size of the population, which is considered as a variable associated with the economies of scale in the provision of public goods, had the expected sign; that is, there was a positive and statistically significant marginal effect of the population on the supply of open government data in all of our estimated models. As we mentioned before, the non-excludable property of open government data means that there could be economies of scale in the costs of providing this pure public good. This means that the per capita costs of providing a pure public good are decreasing as the cost of public goods are shared among more people (i.e., as the size of the population in a country increases). This, in turn, leads to a fall in the per capita cost of providing public goods, which increases the demand for this type of goods. Hence, governments respond by increasing the corresponding supply of such goods.

Our models, through the individual t-test (the t-tests are displayed in Table 1 in parenthesis), also suggest that demographic dummy variables and dummy variables capturing the global distribution of income are not statistically significant to explain the cross-country differences in the supply of open government data in our sample (see Models III, IV, and V). However, the F statistic shows that, jointly, all independent variables considered in Models I–V are statistically significant to explain the cross-country differences in the supply of open government data in our sample. To see this, we tested if $\beta_1 = \beta_2 = \beta_3 = \beta_4 = \beta_5 = \beta_6 = \beta_7 = \beta_8 = \ldots \beta_k = 0$; that is, we tested if the joint effect of our independent variables helped to explain cross-country differences in the supply of open government data, and the corresponding F statistic showed that we rejected the null hypothesis shown above. Therefore, Models I–V as a whole are statistically significant (see Table 1).

## 5. Conclusions

We developed a cross-section analysis to provide tests for the institutional, political, and economic determinants of cross-country differences in the supply of open government data. We consider open government data as a pure public good and, therefore, it satisfies two properties: open government data is a non-excludable good (once open government data is provided, then any person who seeks to have access can have access to that good), and it is a non-rival good (the consumption of the good by some agent does not preclude the consumption of the same good by everyone else). We used an index of the supply of open government data and estimated a cross-section regression model to analyze the cross-country differences in civil rights, transparency, quality of government, the size of the economy, the size of the population, political participation, and sociodemographic characteristics. These can explain cross-country differences in the supply of open government data. We also conducted a robustness check of our analysis by estimating five different models of regression analysis that also include dummy variables associated with geographic heterogeneity and the global distribution of income.

Our analysis provided evidence that cross-country differences in governance and social institutions such as civil liberties, government transparency, and the quality of democracy are statistically significant predictors of cross-country differences in the supply of open government data. Our estimates suggest that civil rights and the transparency of government in each country have a marginal positive and statistically significant effect on the supply of open government data. In addition, our variable that captured changes in the demand of web resources—that is, penetration of users, or the proportion of Internet users over the country's population—was both positive and statistically significant in all of our estimated models. Our analysis also suggests a differentiated response of governments to changes with

the demand of web resources. In particular, if there is an increase in the demand of Internet services, countries with higher qualities of democracy supply more open government data than countries with weaker democracies. This result also shows that the level of governance in each country is an important determinant of the cross-country differences in the supply of open government data.

In our analysis, we included two variables that were associated with the efficiency of government intervention and with economies of scale in the provision of public goods. In this paper, we found evidence that the efficiency of government intervention and the economies of scale can also explain the cross-country differences in open government data. The efficiency of government intervention has a negative marginal effect on open government data. One possible explanation of this outcome is that an increase in the efficiency of government intervention might lead to more resources available to be spent by governments. However, those available resources might be spent on other governmental programs that might have a higher electoral impact relative to the choice of supplying more open government data, thus explaining the negative marginal effect of efficiency on the supply of open government data. In addition, the size of the population in a country was considered as a variable associated with economies of scale in the provision of public goods. Our analysis showed that there is a positive and statistically significant marginal effect of the population on the supply of open government data in all of our estimated models. As we mentioned before, the non-excludable property of open government data means that there could be economies of scale in the cost of providing open government data. This means that the per capita costs of providing a pure public good decrease as the cost of public goods is shared among more people. This, in turn, leads to a fall in the per capita cost of providing public goods, which increases the demand for this type of goods. Hence, governments respond by increasing the corresponding supply of open government data.

In addition, our models provide weak support to the hypothesis that open government data is a normal good; that is to say, countries with higher incomes are associated with higher levels of supply of open government data. However, in our models, political participation, the sociodemographic characteristics of citizens, demographic dummy variables, and dummy variables capturing the global distribution of income did not help to explain cross-country differences of the supply of open government data.

It is relevant to mention that the main limitation of our analysis is that we used cross-section data for our regression analysis, which limited the generality of our results. We decided to use data from the GODI for the year 2016 because this is the most up-to-date data on the GODI. Even if there is data for the Global Open Data Index for other years, the Open Knowledge Foundation has clearly stated that changes in methodology in the calculation of the GODI make unsuitable the comparison of data between 2016 and other years. This limits the study of what factors could explain the changes of the GODI over time. However, this limitation could be eased as long as more data sets become available in the future that allow other forms of regression analysis, such as regression with panel data, that might improve the properties of estimation and hypothesis testing, as well as the generality of the results.

**Author Contributions:** Conceptualization, A.P. and R.A.P.R.; methodology, A.P. and R.A.P.R.; validation, A.P. and R.A.P.R.; formal analysis, A.P. and R.A.P.R.; investigation, A.P. and R.A.P.R.; data curation, A.P. and R.A.P.R.; writing—original draft preparation, A.P. and R.A.P.R.; writing—review and editing, A.P. and R.A.P.R. All authors have read and agreed to the published version of the manuscript.

**Funding:** This research received no external funding.

**Acknowledgments:** Both authors would like to thank Universidad Autónoma de Ciudad Juárez for the institutional support that made this research possible.

**Conflicts of Interest:** The authors declare no conflict of interest.

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
