# Peer review of "An Analysis of the Supply of Open Government Data"

_futureinternet, doi:10.3390/fi12110186_

Round 1

Reviewer 1 Report

Indoor first sentence of the abstract you assert three aspects which raise questions that are not answered.

Whilst data science can be used as a tool, the question of the variance in open data supply cannot be answered by data analyses alone. To answer such a question requires a thorough qualitative approach, as aspects such as the governance of open data in each country and which governance instruments, such as collective decision-making structures, strategic management, allocation of tasks and responsibilities, creation of markets, interorganizational culture and knowledge management, and regulation and formalization of open data initiatives are in place. In addition, the open data maturity level (data readiness, data quality and data impact) of the various countries have to be taken into account. 

The GODI index has not been actualised since 2016, so your article only provides a snapshot of 2016. You need to make this clear in your introduction. GODI does not assess the quality of the data, only if the data are published according to open data principles. In addition, GODI only assessed datasets published at national level, whereas a number of the assessed datasets are published by governments at local level. So, the data are available as open data, just not harvested (in 2016) by national catalogue services. In your analysis, have you analysed all 94 places (not countries!) assessed in GODI? To which extent is your method reproducible, so a follow-up study could be carried out, say after the implementation of the 2019/1024/EU Open Data Directive, to assess developments since 2016.

You need to be more transparent about the assessment criteria / indicators / determinants you used. In addition, you need to describe the limitations of your research in more detail. In both your introduction and in your conclusion, you confuse cause and effect. 

Please see attached file for comments in more detail.  

Author Response

To whom it may concern  

In this document, you will find the updated version of our paper “An Analysis of the Supply of Open Government Data”. First, we would like to thank you for your valuable comments. We are glad to report that this new updated version takes into consideration all the comments and suggestions from you, and in this document, we explain in detail the changes made to the paper-based on comments and suggestions. 

Best regards. 

Alan Ponce  

Raúl Alberto Ponce Rodríguez. 

Reviewer 2 Report

This is a timely topic. It's exciting to read a paper that tries to explain the differences in the OGD supply. The introduction section is very strong, and overall this is a well-designed and well-executed study.  However, I think section 2 and section 3, especially section 3, can (and should) be improved.

Section 2.1 Determinants of the supply of OGD:
Based on a few publications, the authors’ discussed the possible determinants of the supply of OGD. I think this section needs more content about the literature on the public good and how that literature informs this research. Authors’ own reasoning and discussion are important for this research but seem a little out of place or not well supported since this section is framed around the provision of public goods. The entire section 2.1 only cites four references. The paragraph starting at line 169 and the one starting at line 176 do not cite any literature. Line 202, “In addition, most theoretical models that study the provision of pure public goods consider the size of population as an important determinant of the provision of pure public goods.” Here I think citations are necessary.

Section 3:
The authors probably want to explain the variables better, as well as the data sources selected. If I understand it correctly, there are nine variables, listed on lines 294-303. These are selected based on the determinants postulated in section 2.1. Some of the variables are explained or self-explanatory, but some are not, such as “the effect of the quality of democracy of a country” and “the efficiency of the government.” The definitions are essential to the research and should decide what data sources to use. I think all these elements--variables, definitions of the variables, and data sources—are worth more explanation and should follow immediately after the discussion of the determinants, that is, at the beginning of section 3. I don’t understand the logic behind the current structure of section 3—introducing the data sources before the variables. Also, could the authors explain lines 260-262 “In order to complete a full sample with the desired variables for our empirical analysis, our final cross-sectional dataset is composed of 49 observations and 66 variables?” I suppose the desired variables are those listed on lines 294-303, but what about the 66 variables? How these 66 variables are used?

Again, overall I think the research is strong, but revisions are necessary to make the method more clear.

Author Response

(The authors gave the same response as above.)

Round 2

Reviewer 1 Report

Although the authors have addressed a number of issues indicated in the first version, other issues have emerged. The section on positive literature (p.5-6) does not, in my opinion, add much to explaining the correlation between open government data and elections. The term 'imperfect electoral competition' is not explained. The literature the section is based on, all predates the concept op open government data (OGD) as a public good, and therefore, the relevance of OGD to imperfect electoral competition.  This whole section appears to ignore the role of open government data, i.e. providing information to citizens.  The authors seem to treat OGD as if it were the same as any other public good, such as public transport. This is demonstrated by the fact that in this revised version, the authors still consider 'the public sector information' (line 149) to be an initiative(!)

The latter part of the introduction of the article (from line 71 onwards) is more of a summary than providing more background on the research question. 

your conclusion is interesting. There is a correlation between countries with a higher level of civil liberties, government transparency and the quality of democracy and the supply of open government data. Our gut feeling indicated this as well bu now there is statistical evidence that this was the case in 2016. 

There are still some textual issues, such as missing commas, incorrect use of definitive articles, missing spaces, long sentences, etcetera. 

For more details, see attached document

Author Response

To whom it may concern 

In this document, you will find the updated version of our paper “An Analysis of the Supply of Open Government Data”. First, we would like to thank our reviewer for valuable comments. We are glad to report that this new updated version takes into consideration all the comments and suggestions from you, and in this document, we explain in detail the changes made to the paper-based on comments and suggestions.

Best regards.

Alan Ponce 

Raúl Alberto Ponce Rodríguez.
